# Adaptive deep spiking neural network with global-local learning via balanced excitatory and inhibitory mechanism

**Tingting Jiang[1], Qi Xu[1]***, **Xuming Ran[1], Jiangrong Shen[2], Pan Lv[2], Qiang Zhang[1], Gang Pan[2]**
[1]School of Computer Science and Technology, Dalian University of Technology
[2]College of Computer Science and Technology, Zhejiang University
`{jiangtt,xuqi,zhq}@dlut.edu.cn, ranxuming@gmail.com,`
`{jrshen, lvp, gpan}@zju.edu.cn`

## Abstract

The training method of Spiking Neural Networks (SNNs) is an essential problem, and how to integrate local and global learning is a worthy research interest. However, the current integration methods do not consider the network conditions suitable for local and global learning and thus fail to balance their advantages. In this paper, we propose an Excitation-Inhibition Mechanism-assisted Hybrid Learning (EIHL) algorithm that adjusts the network connectivity by using the excitation-inhibition mechanism and then switches between local and global learning according to the network connectivity. The experimental results on CIFAR10/100 and DVS-CIFAR10 demonstrate that the EIHL not only obtains better accuracy performance than other methods but also has excellent sparsity advantage. Especially, the Spiking VGG11 is trained by EIHL, STBP, and STDP on DVS_CIFAR10, respectively. The accuracy of the Spiking VGG11 model with EIHL is **62.45%**, which is **4.35%** higher than STBP and **11.40%** higher than STDP. Furthermore, the sparsity achieves **18.74%**, which is quite higher than the above two non-sparse methods. Moreover, the excitation-inhibition mechanism used in our method also offers a new perspective on the field of SNN learning.

## 1 Introduction

Spiking neural networks (SNNs) Maass (1997) are a type of neural networks that more closely mimics biological neural systems, with efficiency and energy-saving advantages Xu et al. (2023) Liu et al. (2020) Xu et al. (2020) Xu et al. (2024). The learning algorithm of SNNs is vital to their performance and application, which are mainly divided into two categories: *local learning* Hebb (2005) Song et al. (2000) Lu & Sengupta (2023) and *global learning* Zenke & Ganguli (2018) Wu et al. (2018). *Local learning* is a correlation-driven learning method, suitable for sparse networks and low-power hardware but difficult to handle complex tasks and deep networks. *Global learning* is a loss-driven learning method, suitable for dense networks to solve complex tasks, but requires a lot of computational energy consumption. Therefore, integrating local and global learning so that the learning algorithm has both the energy-saving advantages of local learning and the high-performance advantages of global learning is a significant research interest in the field of SNN.

The biological three-factor learning rules Gerstner et al. (2018) Bailey et al. (2000) provide biological insights for the new way of integrating local and global learning. The three-factor rule is that the adjustment of synaptic weights depends not only on the activity of the pre-synaptic neuron and the post-synaptic neuron but also on a third factor (e.g., neurotransmitter). Wu et al. (2022) also proposed a hybrid synergic learning algorithm, which uses two sets of weights to update the weights of local learning and global learning separately and then directly adds the two sets of weights, achieving comparable performance. However, the current hybrid learning algorithm needs to trade-off between the accuracy advantage of global learning and the low energy consumption advantage of

---

*Corresponding author: Qi Xu, xuqi@dlut.edu.cn

local learning. Moreover, the type of hybrid learning algorithms still needs more reasonable biological plausibility. Hence, the current hybrid learning algorithms have some theoretical and practical shortcomings and need further improvement and optimization via the excitation-inhibition mechanism.

In this paper, we propose an Excitation-Inhibition Mechanism assisted hybrid Learning(EIHL) algorithm. Inspired by the excitation-inhibition mechanism of biological neural networks Rosenberg et al. (2023) Sadeh & Clopath (2021) Wang et al. (2022) Xue et al. (2014) Shea (2021) Simeone & Rho (2009) Wang (2020), We choose to distinguish between excitation and inhibition states according to the overall connection state of the network. When the neural network is in an over-excited state, it switches to an inhibited state. When the neural network is in an over-inhibited state, it switches to an excited state. Based on the dynamic balance of excitation and inhibition in the neural network, we achieve dynamic switching between global learning and local learning.

The main contributions of this paper are as follows:

- We propose an excitation-inhibition mechanism-assisted hybrid learning(called EIHL), which can combine the high accuracy of global learning and the low energy consumption of local learning and showed excellent accuracy and sparsity in the experiments on three datasets and two models.

- We obtained inspiration from neuroscience and adopted the excitation-inhibition mechanism to solve the problem of how to reasonably integrate hybrid learning. The experimental results showed that EIHL also achieved sparsity advantages, which made it possible to deploy on hardware.

- We used the neural excitation-inhibition mechanism to achieve the integration of global and local learning and implemented it by adjusting the weights. This not only provides a new perspective for the field of SNN training methods but also prepares for the generalization of EIHL to the ANN domain.

## 2 RELATED WORKS

### 2.1 LEARNING ALGORITHMS FOR SPIKING NEURAL NETWORKS

*Local learning*: The Hebb learning rule Hebb (2005) states that the pre-synaptic neuron must be activated before the post-synaptic neuron. Based on these rules, spike-timing dependent plasticity (STDP) was proposed by Song et al. (2000). STDP is a local learning algorithm that adjusts synaptic weights based on the pulse timing interval between pre-synaptic and post-synaptic neurons. Deep-STDP Lu & Sengupta (2023) is a local learning algorithm based on STDP, which uses pseudo-labels obtained through clustering for loss backpropagation and updating of synaptic weights, significantly improving performance. However, Deep-STDP also increases computational complexity and energy consumption.

*Global learning*: The Backpropagation (BP) BP (1990) algorithm performs well in artificial neural networks (ANNs) Agatonovic-Kustrin & Beresford (2000), but cannot be directly applied to spiking neural networks (SNNs) Maass (1997). So, the SuperSpike Zenke & Ganguli (2018) algorithm employs a gradient approximation method but does not consider the unique spatiotemporal information of SNNs. The Spatio-Temporal Backpropagation (STBP) Wu et al. (2018) algorithm compensates for the insufficient SuperSpike algorithm. STBP is a modified variant of the backpropagation algorithm specifically designed for training SNNs. STBP considers the temporal relationships and connectivity between spiking neurons, emphasizing the temporal nature of spikes. While STBP performs excellently in task recognition accuracy, it also brings greater computational complexity.

In practice, Wu et al. (2022)proposed hybrid plasticity (HP), which uses two sets of weights to update local and global learning weights, respectively. This hybrid learning method integrates local and global learning, resulting in higher accuracy. HP inspired our method to fuse local and global learning from the weights perspective. In biology, the three-factor learning rules Gerstner et al. (2018) Bailey et al. (2000) indicate that another factor affects synaptic strength besides the activity between presynaptic and postsynaptic neurons, such as neurotransmitters.

## 2.2 EXCITATORY AND INHIBITORY MECHANISMS IN NEURAL SYSTEMS

Neuronal excitation and inhibition mechanisms have pathological and functional significance Rosenberg et al. (2023) Sadeh & Clopath (2021) Wang et al. (2022) Xue et al. (2014). From a pathological perspective, excitation-inhibition not only regulates the emotions of impulse or hesitation but also induces or reduces the frequency and duration of epileptic seizures, as well as improves cognitive and memory functions Rosenberg et al. (2023). From a functional perspective, networks with excitation and inhibition have stronger learning abilities than networks with only excitation, as shown by Wang et al. (2022) who set positive weights as excitation and negative weights as inhibition in an ANN with a monotonic activation function. They proved that networks with only excitation can achieve monotonic functions and can not implement the XOR function.

## 3 PRELIMINARY

### 3.1 INTEGRATE-AND-FIRE MODEL

This paper uses the integrate-and-fire (IF) model as a single computational unit in SNNs, which is a biological neuron model that describes how neurons produce action potentials. The membrane potential of an IF neuron at a certain time step is the decay of the membrane potential from the previous time step plus the external stimulus at this moment. When the membrane potential exceeds a threshold, neuron triggers to firing a spike at this time step. The calculation formula of IF is as follows:

$$\mathcal{M}(t) = \mathcal{V}(t-1) + \mathcal{I}(t), \tag{1}$$

where $\mathcal{M}(t)$ and $\mathcal{V}(t-1)$ is the membrane potential of the neuron at time step $t$ and $t-1$ without spike firing, respectivley. $\mathcal{I}(t)$ is the external stimulus the neuron receives at time step $t$.

$$\mathcal{S}(t) = \begin{cases} 1 & \text{if } \mathcal{M}(t) \geq \mathcal{V}_{\text{thr}} \\ 0 & \text{if } \mathcal{M}(t) < \mathcal{V}_{\text{thr}} \end{cases} \tag{2}$$

$\mathcal{S}(t)$ is the marker of whether the neuron fires a spike at time step $t$, and $\mathcal{V}_{\text{thr}}$ is the threshold for spike firing. If the membrane potential is greater than the $\mathcal{V}_{\text{thr}}$, it fires; if the membrane potential is less than the $\mathcal{V}_{\text{thr}}$, it does not fire.

$$\mathcal{V}(t) = \mathcal{M}(t) - \mathcal{S}(t) \cdot \mathcal{V}_{\text{thr}} \tag{3}$$

$\mathcal{V}(t)$ is the membrane potential after spike firing. It is obtained by subtracting $\mathcal{V}_{\text{thr}}$ from $\mathcal{M}(t)$ if there is spike firing.

### 3.2 SPIKE-TIMING-DEPENDENT PLASTICITY

Spike-Timing Dependent Plasticity (STDP) Song et al. (2000) is a synaptic plasticity mechanism that depends on the firing sequence of pre- and post-synaptic neurons. In STDP, if the pre-synaptic neuron fires before the post-synaptic neuron, synaptic strength increases; if the pre-synaptic neuron fires after the post-synaptic neuron, synaptic strength decreases. Long-Term Potentiation (LTP) Malenka et al. (1999a) and Long-Term Depression (LTD) Ito (1989) are the two components that constitute STDP Zenke et al. (2017). However, significant LTP only occurs at synapses with relatively low initial strength, whereas the extent of LTD does not show an obvious dependence on the initial synaptic strength Bi & Poo (1998). Therefore, the overall effect of STDP tends to exhibit the LTD.

$$\Delta\omega_{ij} = \begin{cases} \mathcal{A} \cdot \exp\left(-\frac{|t_i - t_j|}{\tau_+}\right) & \text{if } t_i \leq t_j, \ \mathcal{A} > 0 \\ \mathcal{B} \cdot \exp\left(-\frac{|t_i - t_j|}{\tau_-}\right) & \text{if } t_i > t_j, \ \mathcal{B} < 0 \end{cases} \tag{4}$$

where $\mathcal{A}$ and $\mathcal{B}$ are the maximum values of synaptic weight change, $\tau_+$ and $\tau_-$ are time constants. STDP determines the sign of weight change $\Delta\omega_{ij}$ based on the order of spike firing of pre-neuron $i$ and post-neuron $j$, and the magnitude of $\Delta\omega_{ij}$ based on the time interval between pre-spike and post-spike firing.

### 3.3 SPATIO-TEMPORAL BACKPROPAGATION

Spatio-temporal backpropagation (STBP) Wu et al. (2018) combines the elements of STDP and backpropagation (BP) for training SNNs. In STBP, the STDP rule is usually used to adjust the synaptic weights between neurons, simulating the biological learning rule. Then, the BP algorithm is used to adjust the parameters in the artificial neural network. Due to BP's powerful learning capabilities, most weights increase after STBP learning, which corresponds to the excitation of LTP. Eq. 5 simply introduces the BP loss function $L$ in STBP.

$$\mathcal{L} = \frac{1}{2\mathcal{G}} \sum_{g=1}^{\mathcal{G}} \left\| \mathcal{Y}_g - \frac{1}{\mathcal{T}} \sum_{t=1}^{\mathcal{T}} \mathcal{O}_g^{t,N} \right\|_2^2, \tag{5}$$

where $\mathcal{G}$ is the number of samples in a batch, $\mathcal{Y}_g$ is the label of the $g$ sample in a batch, $\mathcal{T}$ is the size of the time window. $\mathcal{O}_g^{t,N}$ is the model output of the $g$ sample at time step $t$, where $N$ is the number of classes. It represents the prediction of the model for the $g$ sample at time step $t$.

## 4 EXCITATION-INHIBITION MECHANISM-ASSISTED HYBRID LEARNING

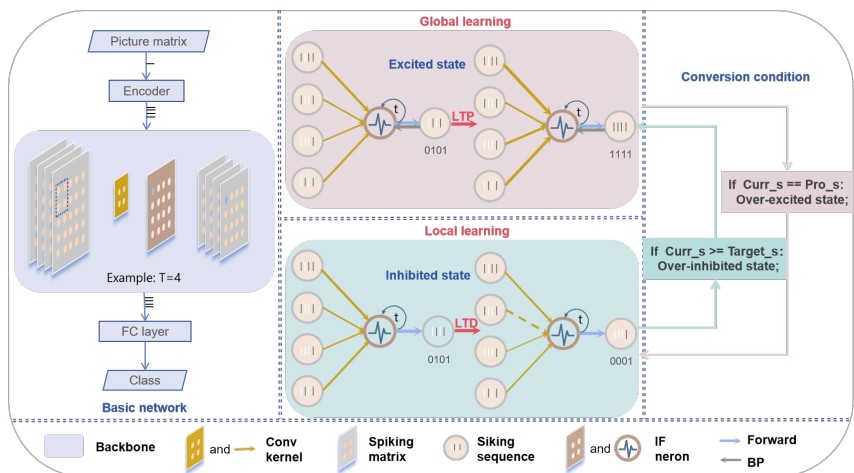

Figure 1: The illustration of the training processing SNNs via EIHL. The solid lines represent the connections that exist, and the dashed lines represent the connections that are broken. In global learning, the LTP effect of excitation is synaptic enhancement, which makes the neurons in the network more active to fire spikes. In local learning, the LTD effect of inhibition is synaptic shrinkage or even disconnection, which makes the neurons in the network more passive to fire spikes.

First of all, the challenge of hybrid learning lies in the integration of global and local learning. While current hybrid learning methods have achieved excellent results in terms of accuracy Wu et al. (2022), they could do slightly better in integrating the low energy consumption advantage of local learning. Therefore, based on the exciting Long-Term Potentiation (LTP) Malenka et al. (1999a) results of global learning and the inhibitory Long-Term Depression (LTD) Ito (1989) results of local learning, an excitatory-inhibitory mechanism is adopted to balance the two. Secondly, in the cerebral cortex, the excitatory mechanism can enhance synaptic strength Malenka et al. (1999b), while the inhibitory mechanism can weaken it Abraham & Bear (1996) Li et al. (2017b). Unlike previous works Kern & Chao (2023) Zhu et al. (2017) that directly distinguish between excitation and inhibition at the synaptic and neuronal levels from a microscopic perspective, we use the network connection status to differentiate between excitatory and inhibitory states from a macroscopic perspective. Finally, according to the excitatory-inhibitory mechanism, excitation and inhibition are automatically balanced. As shown in Fig. 1. when the network is overly excited, it should be inhibited, and when it is in an overly inhibited state, it should be excited. We propose an Excitatory-Inhibitory Hybrid Learning (EIHL) method, which better integrates the advantages of local and global learning, resulting in a model with high accuracy and low power consumption. We use the

---

**Algorithm 1:** Excitation-Inhibition Mechanism assisted Fusion learning method.

---

**Require:** current sparsity $Curr\_S$, previous sparsity $Pre\_S$, current target sparsity $b$, the increment step of parameter $b\ step$.
**Ensure:** SNN model with EIHL
    **for** i=1 to epoch **do**
      **if** $Curr\_S\ >=\ Pre\_S\ and\ Curr\_S\ <\ b$ **then**
        Running STDP;
        Disconnect the weak connections of excitation or inhibition according to the LTD
        principle using Eq. 6 and Eq. 7;
      **else**
        Running STBP and $x$ undergoes self-decay in Eq. 6.;
        **if** just flipped to STBP **then**
          $b \leftarrow b + step$
        **end if**
      **end if**
    **end for**

---

degree of network connectivity to distinguish between excitation and inhibition, so we set a target sparsity $b$ as the threshold for the network to be in an over-inhibited state. In local learning, due to the effect of LTD, the network sparsity gradually increases. When it exceeds the target sparsity $b$, the network is considered to be in an over-inhibited state, and the network will switch to excited global learning. The contraction formula Eq. 6 accelerates the process of the network gradually becoming sparse in local learning.

$$\mathcal{H}(x) = thresh(b) \cdot x \cdot \alpha, \tag{6}$$

Actually, $\mathcal{H}(x)$ is a weight value, and contraction implies that weights less than the boundary value $\mathcal{H}(x)$ will shrink until they disconnect. Moreover, the target sparsity $b$ is a percentage, not a weight value, so $thresh(b)$ is the mapping of b on the weight distribution. In simpler terms, if the weights that are less than $thresh(b)$ in each layer are set to 0, the current sparsity will directly reach the target sparsity $b$. The network will immediately exhibit an over-inhibited state and will switch to excited global learning. However, defining the range and setting it to 0 directly is too crude and will cause a lot of unnecessary losses. Therefore, $x \cdot \alpha$ is to give $\mathcal{H}(x)$ a slow expansion process from 0. $\alpha$ first divides the $thresh(b)$ into multiple scales, and as $x$ gradually increases, $\mathcal{H}(x)$ will also slowly increase until it equals $thresh(b)$.

However, we believe that simply setting the weights below the boundary value $\mathcal{H}(x)$ to 0 is still crude. Therefore, we fix the update direction of the weights at the boundary value $\mathcal{H}(x)$ to move only towards 0 and then set the weights near zero to 0, to accelerate LTD in a relatively smooth manner.

$$\mathcal{W}' = \mathcal{W} - \text{lr} \cdot \nabla\mathcal{W}, \tag{7}$$

$\mathcal{W}$ denotes the weight that has not been updated, $\mathcal{W}'$ denotes the weight that has been updated, $\nabla W$ is the weight update amount of STDP and lr denotes the learning rate.

Summarize the contraction operation of EIHL on local learning, that is, The update direction of the weights within the $\mathcal{H}(x)$ boundary can only tend to 0, and the weight area close to 0 can be directly set to 0. Then $x + +$, $\mathcal{H}(x)$ gradually expands, and the final set to 0 area reaches the $thesh(b)$ area, the current sparsity reaches the target sparsity $b$, and switches to the excited global learning.

We set the threshold for over-inhibition as $b$, but we did not specify a value for the threshold of over-excitation. The criterion for judging the state of over-excitation is "$Curr\_S \geq Pre\_S$". We assume that in global learning, the network is over-excited when the sparsity stops decreasing, and then switches to inhibited global learning. Meanwhile, in order to better demonstrate the low-power advantage of local learning, the threshold for over-inhibition $b$ is incremented by a step size every time it switches.

## 5    EXPERIMENTAL SETTINGS

### 5.1    DATASETS

In this section, we conduct experiments with two models: Spiking Resnet18 res Han et al. (2020) and Spiking VGG11 Sengupta et al. (2019b), on three datasets: CIFAR10, CIFAR100 Krizhevsky et al. (2009), and DVS-CIFAR10 Li et al. (2017a). We compare our methods with local learning methods (STDP) and global learning methods (STBP) to demonstrate their superiority in accuracy, robustness, and network sparsity.

**CIFAR10** is a small RGB dataset with 10 classes. Each category has 6000 images, of which 5000 are used for training and 1000 are used for testing. The shape of each image is $32\times32\times3$. It has some representativeness and universality. It can verify whether the obtained model has general classification ability.

**CIFAR100** is an extension of CIFAR10, with 20 superclasses, each containing 5 classes. Each category has 600 images, of which 500 are used for training and 100 are used for testing. The shape of each image is also $32\times32\times3$. It is a very challenging classification task.

**DVS-CIFAR10** is an event-based dataset for object classification, which is obtained by scanning the CIFAR10 dataset with a Dynamic Vision Sensor (DVS) Gallego et al. (2022). It has a total of 10,000 event streams, of which 8,000 are used for training and 2,000 are used for testing. The shape of each event stream is $128\times128\times2$, where the last dimension represents the timestamp and polarity. It is more suitable for the training and testing of Spiking Neural Networks.

### 5.2    PATTERN RECOGNITION TASK

We conducted four experiments to evaluate our method. First, we tested our method on CI-FAR10/100 and DVS_CIFAR10, and compared the accuracy and sparsity with local learning STDP and global learning STBP, to verify the essential performance of the model obtained by EIHL. Secondly, we supplemented an ablation study experiment on the increment step of parameter $b$ in the contraction curve Eq. 6 that directly affects the sparsity in the EIHL method on CIFAR10, and controlled the sparsity of the network by the step size, to test the sparsity advantage of the model obtained by EIHL. Thirdly, we did a random pruning experiment on CIFAR10, which is to randomly prune 20%, 40%, and 60% of the network connections, and test the capability of the network to classify spatiotemporal patterns under the EIHL method. Finally, we compare the performance of EIHL and other hybrid learning methods on CIFAR10, to verify the ability of EIHL. Furthermore, for all the experiments, we used Spiking Resnet18 and Spiking VGG11 to verify the generalization ability of our method. In these experiments, the sparsity is an additional advantage for our EIHL method. The sparsity evaluation metric computes the proportion of weights that are 0 among all the weight parameters. The accuracy evaluation metric equals the proportion of samples correctly classified in the test set.

### 5.3    IMPLEMENT DETAILS

We conduct experiments on a server equipped with a 16-core Intel(R) Xeon(R) Xeon(R) Gold 6330 2.80GHz CPU and 20 NVidia GeForce RTX 3090 Ti GPUs. In these experiments, we used the SpikingJelly framework Fang et al. (2020) to simulate the whole training process, the parameters of the STDP layer were updated by the SGD optimizer Mandt et al. (2017), and the global information was updated by the Adam optimizer Zhang (2018). The two optimizers shared a learning rate. The learning rates of Resnet18 on CIFAR10, CIFAR100 and DVS_datasets are 2e-3, 5e-4 and 2e-3 respectively. The learning rates of VGG11 on three datasets are 2e-4, 2e-4 and 1e-5 respectively. Besides, the increment step of parameter $b$ in the experiments of Tab. 1 and Tab. 3 is 0.5. The number of epochs in the four experiments is 200.

## 6    EXPERIMENTAL RESULTS

Firstly, we evaluate the sparsity and accuracy of EIHL's experimental results as shown in Tab. 1. EIHL achieved certain accuracy advantages and ensured an advantageous sparsity. Secondly, we

investigate the influence of the contraction boundary value in EIHL's experimental results as shown in Tab. 2. We found that, overall, the step size is proportional to the sparsity and inversely proportional to the accuracy. Thirdly, we compare the performance of the model on different disconnect degrees as shown in Tab. 3. EIHL still maintained a certain accuracy and high sparsity under different degrees of random pruning. Finally, we compare the performance of the model with other hybrid learning methods as shown in Tab. 4. The experimental results show that the EIHL method still has superior accuracy and unique sparsity advantages.

## 6.1 EVALUATION THE SPARSITY AND ACCURACY OF EIHL

We first tested the spiking resnet18 model on three datasets, and it outperformed STBP and STDP in terms of accuracy as well as sparsity. For example, on the CIFAR100 dataset, EIHL's accuracy is 0.15% higher than STBP and 25.97% higher than STDP. Moreover, EIHL's sparsity is 31.75% higher than the other two pure global or local methods. Then, we used another model framework - Spiking VGG11 to evaluate the generalization ability of EIHL. The results show that the test accuracy on all datasets is higher than that of STBP and STDP alone. And it also retains a certain sparsity. As shown in Fig. 2, EIHL's accuracy was almost always surpass STBP in the later about 25 epoches stages. It has been shown that when the current sparsity reaches the target sparsity in the EIHL approach, the target sparsity increases by one step when entering STBP, whereas the current sparsity steadily decreases. When the current sparseness no longer shifts, it will switch to STDP during learning, and the current sparsity will steadily grow until it approaches the desired sparsity. Although the EIHL accuracy is approximately equal to STBP on the VGG model and CIFAR10 dataset, it is because the increment step of $b$ is 0.5 here. When the increment step of $b$ is 2 in Tab. 2, the EIHL method has a slightly higher accuracy than STBP, and also a 28.42% higher sparsity. Moreover, when the increment step of $b$ is 0.125 in Tab. 2, the EIHL accuracy is 0.3% higher than STBP, and also 4.75% higher in sparsity.

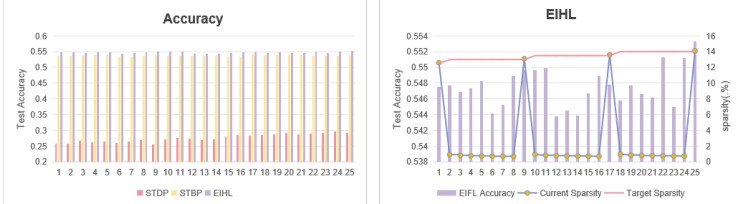

Figure 2: This is the curve variation of the precision and sparseness of the final 25 epochs. **Left:** The comparison of the accuracy of three methods on CIFAR100 using the VGG11 model. It can be seen that the accuracy of EIHL is significantly higher than that of STDP and STBP. **Right:** The change in current sparsity and target sparsity in EIHL. It can be seen that in the EIHL method, after the current sparsity reaches the target sparsity, the target sparsity will increase by one step when entering STBP, and the current sparsity will gradually decrease. When the current sparsity no longer changes, it will flip to STDP for learning, and the current sparsity will gradually increase until it reaches the target sparsity at this time.

Table 1: Evaluation of sparsity and accuracy on CIFAR10/100 and DVS_CIFAR10 datasets between the EIHL and other methods.

| | Spiking Resnet18 Hu et al. (2021) | | | | | |
| | CIFAR10 | | CIFAR100 | | DVS_CIFAR10 | |
| Model | Sparsity.(%) | Accuracy.(%) | Sparsity.(%) | Accuracy.(%) | Sparsity.(%) | Accuracy.(%) |
|---|---|---|---|---|---|---|
| STDP | 0.00 | 76.71 | 0.00 | 32.66 | 0.00 | 36.00 |
| STBP | 0.00 | 89.53 | 0.00 | 58.48 | 0.00 | 60.2 |
| EIHL | 17.17 | 90.25 | 31.75 | 58.63 | 11.42 | 62.9 |
| | Spiking VGG11 Sengupta et al. (2019b) | | | | | |
| | CIFAR10 | | CIFAR100 | | DVS_CIFAR10 | |
| STDP | 0.00 | 78.29 | 0.00 | 29.17 | 0.00 | 51.05 |
| STBP | 0.00 | 85.76 | 0.00 | 53.75 | 0.00 | 58.10 |
| EIHL | 13.10 | 85.75 | 14.12 | 55.33 | 18.74 | 62.45 |

Therefore, the proposed hybrid learning methods of EIHL achieves superior performance on accuracy and sparsity compared with pure local or global learning mechanisms for SNNs models. At the same time, compared with the hybrid plasticity (HP) method proposed in Wu et al. (2022), with a sparsity of 0 our EIHL method has a certain sparsity advantage.

## 6.2 THE INFLUENCE OF THE CONTRACTION BOUNDARY VALUE IN EIHL

As shown in the right Fig. 2, the current sparsity increases with the gradual increase of $b$. Therefore, in the second experiment, we set different increment steps of $b$ to control the sparsity of the network. The experimental results are shown in Tab. 2. When $step = 0.125$, the accuracy of Resnet18 is 90.00%, and the accuracy of VGG11 is 86.06%, which is still higher than the accuracy of STBP and STDP. However, because the step size is small, the final sparsity of the network is 6.65% and 4.75%. When $step = 4$, the accuracy of the two models is 74.53% and 84.66%, and the sparsity is 80.77% and 48.68%. Too large a step size will cause some damage to the accuracy of the model. It can be seen that the step size of b is basically proportional to the sparsity and inversely proportional to the accuracy.

Although the accuracy is the highest for the Resnet18 model and CIFAR10 dataset when the increment step $b$ is 0.5, the accuracy changes are not significant between the steps of 0.125, 0.25, and 0.5, and the change does not exceed 0.25%. Therefore, from an overall perspective, the increment step of b is still proportional to the sparsity and inversely proportional to the accuracy. Moreover, this experiment also shows that the neural excitation-inhibition mechanism also bring an absolute sparsity advantage to EIHL.

Table 2: Evaluation of sparsity and accuracy on CIFAR10 between difference increment step of b.

| | SNN Model | | | |
| | Spiking Resnet18 | | Spiking VGG11 | |
| Value of b | Sparsity.(%) | Accuracy.(%) | Sparsity.(%) | Accuracy.(%) |
| --- | --- | --- | --- | --- |
| 0.125 | 6.65 | 90.00 | 4.75 | 86.06 |
| 0.25 | 11.58 | 90.02 | 8.31 | 85.49 |
| 0.5 | 17.17 | 90.25 | 13.10 | 85.75 |
| 1.000 | 30.31 | 89.61 | 20.28 | 85.65 |
| 2.000 | 44.62 | 89.08 | 28.42 | 85.77 |
| 4.000 | 80.77 | 74.53 | 48.68 | 84.66 |

According to the experimental results, EIHL can control the sparsity of the network structure by the increment step of $b$, and has a stronger sparsity advantage than the separate local learning STDP and the separate global learning STBP.

## 6.3 COMPARISON THE PERFORMANCE OF THE MODEL ON THE DIFFERENT DISCONNECT DEGREE

In the third experiment, we further verified the robustness and sparsity advantage of EIHL by randomly pruning the network with different disconnect degrees. For example, the 20% random pruning experiment is to randomly select 20% of the weights in each layer and set them to 0 at the initial stage. And keep them and their gradients as 0 in the subsequent learning. As illustrated in Tab.3, under 20% and 40% pruning proportion, the accuracy of EIHL still exceeds that of STBP and STDP. For example, under 40% pruning, the accuracy of the Resnet18 model on EIHL is 0.36% higher than STBP and 11.99% higher than STDP, and the sparsity is 29.73% higher than the other two methods. Although under the Resnet18 model, when pruning 60%, the accuracy of EIHL is no longer higher than STBP, that is because the sparsity of EIHL has reached 87.26%, and the accuracy only drops by 0.26% compared with STBP with 60% sparsity.

This experimental result shows that, under the same pruning level, EIHL performs better than the separate local learning STDP and the separate global learning STBP in the aspects of both accuracy and sparsity. It also proves that EIHL has a certain robustness, as the disconnect degrees increase, the EIHL keeps higher accuracies than the other separate methods.

Table 3: Evaluation of sparsity and accuracy on CIFAR10 between different disconnect degrees.

| | Spiking Resnet18 | | | | | |
| | 20% | | 40% | | 60% | |
| Model | Sparsity.(%) | Accuracy.(%) | Sparsity.(%) | Accuracy.(%) | Sparsity.(%) | Accuracy.(%) |
|---|---|---|---|---|---|---|
| STDP | 20.00 | 77.70 | 40.00 | 75.80 | 60.00 | 72.63 |
| STBP | 20.00 | 89.15 | 40.00 | 87.43 | 60.00 | 84.05 |
| EIHL | 45.24 | 89.24 | 69.73 | 87.79 | 87.26 | 83.79 |
| | Spiking VGG11 | | | | | |
| | 20% | | 40% | | 60% | |
| STDP | 20.00 | 74.72 | 40.00 | 72.25 | 60.00 | 69.14 |
| STBP | 20.00 | 85.22 | 40.00 | 84.23 | 60.00 | 82.67 |
| EIHL | 33.64 | 85.62 | 50.10 | 84.36 | 74.67 | 82.93 |

## 6.4 COMPARISON THE PERFORMANCE OF MODEL WITH OTHER HYBRID LEARNING METHODS

In the last experiment, we conducted a performance comparison experiment between EIHL and Excitatory-Inhibitory Cooperative Iterative Learning (EICIL) Shao et al. (2024). EICIL is a hybrid training method that simulates the excitatory and inhibitory behaviors of biological neurons and seamlessly integrates them into the training process of Spiking Neural Networks (SNNs). It also has high-precision performance. EICIL proposes two training methods: the iteration using the Surrogate Gradient Method(GS) method and STDP-BW(which incorporates the backpropagation technique into the STDP model) method as GSI, and the iteration using the GS method and STDP-BW-GS method as GSGI.

Table 4: Evaluation of sparsity and accuracy on CIFAR10 between the EIHL and EICIL.

| | SNN Model | | | |
| | Spiking Resnet18 | | Spiking VGG11 | |
| Value of b | Sparsity.(%) | Accuracy.(%) | Sparsity.(%) | Accuracy.(%) |
|---|---|---|---|---|
| GSI | 0.00 | 89.32 | 0.00 | 85.63 |
| GSGI | 0.00 | 88.95 | 0.00 | 85.66 |
| EIHL | 17.17 | 90.25 | 13.10 | 85.75 |

As shown in Tab. 4, EIHL's accuracy is higher than GSI and GSGI under both models. Especially on the Spiking resnet18 model, EIHL's accuracy is 0.93% higher than GSI and 1.3% higher than GSGI. And compared to GSI and GSGI, our EIHL shows a unique sparsity advantage.

## 7 CONCLUSION

We propose an Excitement-Inhibition Mechanism assisted Hybrid learning method (EIHL) for spiking neural networks (SNNs). Benefitting from the automatic harmonization of the excitement-inhibition mechanism, the hybrid learning method with local and global learning is designed to extend the learning scenario of SNNs, which can automatically adjust the network connection status according to the LTP and LTD principles and then switches between local learning and global learning modes according to different network connection states. Hence, the proposed EIHL method overcomes the drawback of current hybrid learning methods that fail to take advantage of the low energy consumption of local learning and the high accuracy of global learning. Experimental results demonstrated that the SNNs model trained by the proposed EIHL method has better accuracy, certain sparsity, and strong robustness compared with other common-used methods.

As to the future work, the proposed method needs to be evaluated on large-scale datasets Russakovsky et al. (2015) and other network architectures Sengupta et al. (2019a) Mostafa (2017) on GPU and hardware platforms, as EIHL can achieve strong sparsity and have significant potential once deployed on hardware platforms Roy et al. (2019). What's more, other variants of local and global learning methods could be explored to improve the performance of EIHL further Zhang et al. (2021) Lu & Sengupta (2023).

ACKNOWLEDGE

This work is supported by National Key R&D Program of China (2021ZD0109803), National Natural Science Foundation of China (NSFC No.62206037, 62306274), National Science Fund for Distinguished Young Scholars (NSFC No.61925603). The Open Project of Anhui Provincial Key Laboratory of Multimodal Cognitive Computation, Anhui University, No. MMC202104, under Grant No. GML-KF-22-11 and the Fundamental Research Funds for the Central Universities (DUT21RC(3)091).

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
