# OpenReview forum: "Adaptive deep spiking neural network with global-local learning via balanced excitatory and inhibitory mechanism"
_ICLR.cc/2024/Conference — ICLR 2024 poster_

### Official Review · Reviewer_p6P6 · 2023-10-27

**Soundness:** 2 fair
**Presentation:** 1 poor
**Contribution:** 1 poor
**Rating:** 5
**Confidence:** 5

**Summary:**

This paper proposes the Excitation-Inhibition Mechanism-assisted Hybrid Learning (EIHL) algorithm for training spiking neural networks. The algorithm combines the global learning and local learning together, and achieves better results than individual learning rules, as demonstrated through the experiments.

**Strengths:**

The method is inspired by biological mechanisms. Researchers in the SNN community should be encouraged to seek inspiration from neuroscience.

**Weaknesses:**

1.The authors do not provide a clear and comprehensive description of their method in section 4. The review raises several concerns, outlined as follows:

 (i). What does the notation $x$ mean? Is it a global parameter for the entire network? How does it "gradually increase" during the STDP period and "decay by itself" during the STBP period?

 (ii). What is the precise formula for the operation $thresh(\cdot)$? It appears that $b$ remains unchanged during the STDP period based on Alg. 1. Does this imply that $thresh(b)$ remains constant in each STDP period?

 (iii). Could the authors provide further elaboration on Eq. 7? Why is gradient descent necessary for updating the weight in the STDP period? How is dL/dx computed, given that L does not seem to be differentiable with respect to x? Also, why do x and W share the same dimension? The reviewer thinks that it should be a scalar. Additionally, why is the "contraction factor" $a$ needed?

 (iv). Regarding alg. 1, how is the "current sparsity" calculated? Why is "Curr S >= Pre S" required in the algorithm?


2.The contribution of this work is not very clear at the current stage. In the reviewer's understanding, this work aims at improving the current hybrid learning algorithms which "have some theoretical and practical shortcomings and need further improvement". However, there is no theoretical (mathematical) results in the paper and the biological plausibility of the proposed hybrid method is not inadequately clarified. Furthermore, the practical preformance is relatively unsatisfactory:

(i). The improvement based on the global learning rule STBP is minimal. In essence, STBP represents a special case of the proposed EIHL with specifically chosen hyperparameters. Consequently, EIHL can consistently yield slightly superior results to STBP through randomness and careful hyperparameter tuning.

(ii). The performance notably lags behind the latest research. Especially, the SOTA results of DVS_CIFAR10 is 20% better than the proposed method.

(iii). There is no comparison between this work and other hybrid methods regarding accuracy and biological plausibility.

(iv). No experiments on large-scale datasets.

In summary, the motivation and contribution of the paper remain ambiguous. The reviewer perceives this work as merely a combination of two exsiting methods without convincing reasons.

3.This paper is not well-written. Several sentences lack coherence, making the overall presentation disjointed. The presentation of equations is arbitrary and non-standard. The resolution of Fig. 1 is low.

**Questions:**

Could the authors provide a more comprehensive explanation of the excitation-inhibition mechanism and how it is used in this work? Although the authors keep mentioning it, the reviewer cannot understand how excitatory and inhibitory synapses are handled differently and how they are balanced.

---

> ### Author Response · Authors · 2023-11-19
> **Response to Weaknesses Ⅰ(part 1)**
>
> We appreciate the feedback from the reviewer. We hope our subsequent responses have adequately addressed their concerns and questions regarding the paper.
>
> * **The authors do not provide a clear and comprehensive description of their method in section 4.**
>
> *Theoretical analysis:*
>
> Firstly, the challenge of hybrid learning lies in the integration of global and local learning. While current hybrid learning methods have achieved excellent results in terms of accuracy[1], they could do slightly better in integrating the low energy consumption advantage of local learning. Therefore, based on the exciting Long-Term Potentiation (LTP) results of global learning and the inhibitory Long-Term Depression (LTD) results of local learning, an excitatory-inhibitory mechanism is adopted to balance the two.
>
> Secondly, in the cerebral cortex, the excitatory mechanism can enhance synaptic strength[2], while the inhibitory mechanism can weaken it[3,4]. Unlike previous works[5,6] that directly distinguish between excitation and inhibition at the synaptic and neuronal levels, we use the network connection status to differentiate between excitatory and inhibitory states.
>
> Finally, according to the excitatory-inhibitory mechanism, excitation and inhibition are automatically balanced. That is, when the network is overly excited, it should be inhibited, and when it is in an overly inhibited state, it should be excited. We propose an Excitatory-Inhibitory Hybrid Learning (EIHL) method, which better integrates the advantages of local and global learning, resulting in a model with high accuracy and low power consumption.
>
> > [1] Wu Y, Zhao R, Zhu J, et al. Brain-inspired global-local learning incorporated with neuromorphic computing[J]. Nature Communications, 2022, 13(1): 65.
>
> > [2] Malenka R C, Nicoll R A. Long-term potentiation--a decade of progress?[J]. Science, 1999, 285(5435): 1870-1874.
>
> > [3] Abraham W C, Bear M F. Metaplasticity: the plasticity of synaptic plasticity[J]. Trends in neurosciences, 1996, 19(4): 126-130.
>
> > [4] Li X, Steffens D C, Potter G G, et al. Decreased between‐hemisphere connectivity strength and network efficiency in geriatric depression[J]. Human brain mapping, 2017, 38(1): 53-67.
>
> > [5] Kern F B, Chao Z C. Short-term neuronal and synaptic plasticity act in synergy for deviance detection in spiking networks[J]. PLOS Computational Biology, 2023, 19(10): e1011554.
>
> > [6] Zhu G, Zhang Z, Zhang X Y, et al. Diverse Neuron Type Selection for Convolutional Neural Networks[C]//IJCAI. 2017: 3560-3566.

---

> ### Author Response · Authors · 2023-11-19
> **Response to Weaknesses Ⅰ(part 2)**
>
> *Formula explanation:*
>
> We use the degree of network connectivity to distinguish between excitation and inhibition, so we set a target sparsity $b$ as the threshold for the network to be in an over-inhibited state. In local learning, due to the effect of LTD, the network sparsity gradually increases. When it exceeds the target sparsity $b$, the network is considered to be in an over-inhibited state, and the network will switch to excited global learning. The contraction formula Eq.6 accelerates the process of the network gradually becoming sparse in local learning.
>
> $\ \mathcal H(x) = thresh(b) \cdot x \cdot a ,\ \ \ \  a \in(0，1)， x \in \mathbb{N}   \ \ \ Eq.6$
>
> Actually, $\mathcal H(x)$  is a weight value, and contraction implies that weights less than the boundary value $\mathcal H(x)$  will shrink until they disconnect. Moreover, the target sparsity $b$ is a percentage, not a weight value, so $thresh(b)$ is the mapping of b on the weight distribution. In simpler terms, if the weights that are less than $thresh(b)$ in each layer are set to 0, the current sparsity will directly reach the target sparsity $b$. The network will immediately exhibit an over-inhibited state and will switch to excited global learning. However, defining the range and setting it to 0 directly is too crude and will cause a lot of unnecessary losses. Therefore, $x \cdot a$ is to give $\mathcal H(x)$ a slow expansion process from 0. $a$ first divides the $thresh(b)$ into multiple scales, and as $x$ gradually increases, $\mathcal H(x)$ will also slowly increase until it equals $thresh(b)$.
>
> However, we believe that directly setting the weights that are less than the boundary value $\mathcal H(x)$ to 0 is still crude. Therefore, we fix the update direction of the weights in the boundary value $\mathcal H(x)$ to only move towards 0, and then set the weights in the zero neighborhood to 0, to achieve the LTD result in a relatively smooth manner.
>
> $\ \mathcal{W}' = \mathcal{W} - \text{lr} \cdot \nabla \mathcal{W},\ \ \ \  \text{lr} > 0   \ \ \ Eq.7 $
>
> $\mathcal{W}$ denotes the weight that has not been updated, $\mathcal{W}'$ denotes the weight that has been updated, $\nabla W$ is the weight update amount of STDP and $\text{lr}$ denotes the learning rate.
>
> Summarize the contraction operation of EIHL on local learning, that is, The update direction of the weights within the $\mathcal H(x)$ boundary can only tend to 0, and the weight area close to 0 can be directly set to 0. Then $x++$ , $\mathcal H(x)$ gradually expands, and the final set to 0 area reaches the $thesh(b)$ area, the current sparsity reaches the target sparsity $b$, and switches to the excited global learning.

---

> ### Author Response · Authors · 2023-11-19
> **Response to Weaknesses Ⅰ(part 3)**
>
> * **The review raises several concerns, outlined as follows:
> (i). What does the notation $x$ mean? Is it a global parameter for the entire network? How does it "gradually increase" during the STDP period and "decay by itself" during the STBP period?**
>
>   $x$ represents the quantity of weights that need to be reduced. $x$ is a global parameter for the entire network. $x$ in STBP and STDP every certain number of batches respectively $x--$ and $x++$, corresponding to the contraction boundary $\mathcal H(x)$  gradually contracting and expanding.
> * **(ii). What is the precise formula for the operation $thresh(b)$? It appears that $b$ remains unchanged during the STDP period based on Alg. 1. Does this imply that $thresh(b)$ remains constant in each STDP period?**
>
> The target sparsity $b$ is a percentage, not a weight value, so $thresh(b)$ is the mapping of b on the weight distribution. $b$ and $thresh(b)$ remains unchanged during the STDP period.
>
> * **(iii). Could the authors provide further elaboration on Eq. 7? Why is gradient descent necessary for updating the weight in the STDP period? How is dL/dx computed, given that L does not seem to be differentiable with respect to x? Also, why do x and W share the same dimension? The reviewer thinks that it should be a scalar. Additionally, why is the "contraction factor" $a$ needed?**
>
> We are extremely grateful to the reviewer for pointing out the error in Eq.7.
>
> Firstly, we believe that directly setting the weights that are less than the boundary value $\mathcal H(x)$ to 0 is still crude. Therefore, we fix the update direction of the weights in the boundary value $\mathcal H(x)$ to only move towards 0 by the Eq.7, and then set the weights in the zero neighborhood to 0, to achieve the LTD result in a relatively smooth manner.
>
> Secondly, as the reviewer said, the gradient in the gradient descent formula is $\nabla \mathcal{W}$, not $\frac{d \mathcal L} {d x}$, which is a mistake in our writing.
>
> Finally, $x \cdot a$ is to give $\mathcal H(x)$ a slow expansion process from 0. $a$ first divides the $thresh(b)$ into multiple scales, and as $x$ gradually increases, $\mathcal H(x)$ will also slowly increase until it equals $thresh(b)$.
>
> * **(iv). Regarding alg. 1, how is the "current sparsity" calculated? Why is "Curr S >= Pre S" required in the algorithm?The authors do not provide a clear and comprehensive description of their method in section 4.**
>
> Firstly, the current sparsity is equal to the proportion of weights that are 0 among all the weight parameters. This is given in section 5.2.
>
> Secondly, we directly set the threshold for over-inhibition as $b$, but we did not specify a particular value for the threshold of over-excitation. The criterion for judging the state of over-excitation is “Curr S >= Pre S”. We believe that in global learning, when the sparsity no longer decreases, the network presents a state of over-excitation.
>
> Finally, we fully respect the reviewer’s comments and have reorganized section 4. We believe this will enhance the clarity and comprehensiveness of our work. Thank you for your valuable feedback.
>
> *We hope that these revisions can help the reviewers to have a clearer understanding of our paper. If the reviewers have any questions or suggestions, please feel free to tell us, and we will improve them as soon as possible.*

---

> ### Author Response · Authors · 2023-11-19
> **Response to Weaknesses Ⅱ (part 1)**
>
> We appreciate the feedback from the reviewer. We hope our subsequent responses have adequately addressed their concerns and questions regarding the paper.
>
> * **The contribution of this work is not very clear at the current stage. In the reviewer's understanding, this work aims at improving the current hybrid learning algorithms which "have some theoretical and practical shortcomings and need further improvement". However, there is no theoretical (mathematical) results in the paper and the biological plausibility of the proposed hybrid method is not inadequately clarified. Furthermore, the practical preformance is relatively unsatisfactory:**
>
> We apologize for the confusion, and then according to the reviewer’s suggestion, we re-summarized the contributions of this work:
>
> 1. We propose an excitation-inhibition mechanism-assisted hybrid learning(called EIHL), which can combine the high accuracy of global learning and the low energy consumption of local learning and showed excellent accuracy and sparsity in the experiments on three datasets and two models.
> 2. We obtained inspiration from neuroscience and adopted the excitation-inhibition mechanism to solve the problem of how to reasonably integrate hybrid learning. The experimental results showed that EIHL also achieved sparsity advantages, which made it possible to deploy on hardware.
> 3. We used the neural excitation-inhibition mechanism to achieve the integration of global and local learning and implemented it by adjusting the weights. This not only provides a new perspective for the field of SNN training methods but also prepares for the generalization of EIHL to the ANN domain.
>
> * **(i). The improvement based on the global learning rule STBP is minimal. In essence, STBP represents a special case of the proposed EIHL with specifically chosen hyperparameters. Consequently, EIHL can consistently yield slightly superior results to STBP through randomness and careful hyperparameter tuning.**
>
> We appreciate the reviewer’s point, but first of all, our motivation is to propose a hybrid learning method that can well integrate the high-precision advantage of global learning and the low-energy consumption advantage of local learning. According to the experimental results analysis, our proposed EIHL achieved the established goal, with higher accuracy than global learning and higher sparsity than local learning. Instead of simply pursuing the improvement of accuracy.
>
> Secondly, in the neural excitation-inhibition mechanism, it is the minority of inhibitory neurons that inhibit the majority of excitatory neurons, and the inhibitory effect is crucial[1-3]. Therefore, we think that the minority of inhibited local learning in EIHL is also crucial.
>
> Finally, in the Tab.1, especially on the DVS dataset, the accuracy of EIHL is 2.7% and 4.35% higher than STBP on the two models, respectively, which is not a gap that can be achieved by simply tuning parameters. In Tab.3, EIHL has higher sparsity than STBP by 10% or even 30% under higher accuracy than STBP, which is obviously not achievable by simply tuning parameters.
>
> * **(ii). The performance notably lags behind the latest research. Especially, the SOTA result of DVS_CIFAR10 is 20% better than the proposed method.**
>
> We agree with the reviewer’s comments and apologize for that, and we will use more advanced models to achieve higher baseline accuracy in future work. The model used in this paper is simple, which directly affects the quality of the results. In addition, there are many parameters that can be adjusted in a network, such as the optimizer, the number of iterations, the learning rate, the learning rate decay rate, etc. There are too many factors that affect the accuracy of the network. Therefore, we also accept the reviewer’s suggestions, and in future work, we will conduct experiments with a better baseline model and parameters.
>
> > [1] Simeone, & Rho, J. M. (2009). Antiepileptic Drugs: Antiepileptic Drug Mechanisms. In Encyclopedia of Basic Epilepsy Research (pp. 59–66). Elsevier Inc. https://doi.org/10.1016/B978-012373961-2.00160-0
>
> > [2] Shea, Thomas. (2021). An Overview of Studies Demonstrating that ex vivo Neuronal Networks Display Multiple Complex Behaviors: Emergent Properties of Nearest-Neighbor Interactions of Excitatory and Inhibitory Neurons. The Open Neurology Journal. 15. 3-15. 10.2174/1874205X02115010003
>
> > [3] Wang X J. Macroscopic gradients of synaptic excitation and inhibition in the neocortex[J]. Nature Reviews Neuroscience, 2020, 21(3): 169-178.

---

> ### Author Response · Authors · 2023-11-19
> **Response to Weaknesses Ⅱ (part 2)  and Ⅲ**
>
> * **(iii). There is no comparison between this work and other hybrid methods regarding accuracy and biological plausibility.**
>
> We appreciate the reviewer’s comments and have conducted a comparative experiment with other hybrid learning algorithms. Below is a performance comparison experiment between EIHL and Excitatory-Inhibitory Cooperative Iterative Learning (EICIL)[4]. EICIL is a hybrid training method that simulates the excitatory and inhibitory behaviors of biological neurons and seamlessly integrates them into the training process of Spiking Neural Networks (SNNs). EICIL proposes two training methods:  the iteration using the Surrogate Gradient Method(GS） method and STDP-BW(which incorporates the backpropagation technique into the STDP model) method as GSI, and the iteration using the GS method and STDP-BW-GS method as GSGI.
>
> We conducted experiments on CIFAR10 using Spiking Resnet18 and Spiking VGG11, respectively. The number of epochs was set to 200, and the learning rates were 2-2e and 2-3e. This experiment aims to provide a comprehensive comparison between the proposed method and existing techniques.
>
> | Learning |       Model      | Sparsity.(%) | &Accuracy.(%) |     Model     | Sparsity.(%) | &Accuracy.(%) |
> |:--------:|:----------------:|:------------:|---------------|:-------------:|:------------:|---------------|
> |   EIHL   | Spiking Resnet18 |     17.17    |     90.25     | Spiking VGG11 |     13.10    |     85.75     |
> |    GSI   | Spiking Resnet18 |     0.00     |     89.32     | Spiking VGG11 |     0.00     |          85.63    |
> |   GSGI   | Spiking Resnet18 |     0.00     |         88.95      | Spiking VGG11 |     0.00     |       85.66        |
>
> The experimental results show that the EIHL method still has superior accuracy and unique sparsity advantages.
>
> * **(iv). No experiments on large-scale datasets.**
>
> We appreciate the reviewer’s comments. We are trying to experiment on large-scale datasets, but the rebuttal time is limited, we hope the reviewer understands.
>
> * **In summary, the motivation and contribution of the paper remain ambiguous. The reviewer perceives this work as merely a combination of two exsiting methods without convincing reasons.**
>
> First of all, the motivation of this paper is to reasonably integrate local and global learning, and to be compatible with the advantages of low power consumption of local learning and high accuracy of global learning.
>
> Secondly, the contribution of this paper is:
>
> 1. We propose an excitation-inhibition mechanism-assisted hybrid learning(called EIHL), which can combine the high accuracy of global learning and the low energy consumption of local learning and showed excellent accuracy and sparsity in the experiments on three datasets and two models.
> 2. We obtained inspiration from neuroscience and adopted the excitation-inhibition mechanism to solve the problem of how to reasonably integrate hybrid learning. The experimental results showed that EIHL also achieved sparsity advantages, which made it possible to deploy on hardware.
> 3. We used the neural excitation-inhibition mechanism to achieve the integration of global and local learning and implemented it by adjusting the weights. This not only provides a new perspective for the field of SNN training methods but also prepares for the generalization of EIHL to the ANN domain.
>
> Finally, the purpose of this paper is to explore how to integrate local learning and global learning, rather than to create two new learning algorithms.
>
> * **This paper is not well-written. Several sentences lack coherence, making the overall presentation disjointed. The presentation of equations is arbitrary and non-standard. The resolution of Fig. 1 is low.**
>
> We have carefully checked and polished the whole paper, and we have tried our best to make the paper more readable and coherent. Then, we also re-optimized our figures and formulas to present our work more clearly.
>
> *We hope that these revisions can help the reviewers to have a clearer understanding of our paper. If the reviewers have any questions or suggestions, please feel free to tell us, and we will improve them as soon as possible.*
>
> > [4] Shao Z, Fang X, Li Y, et al. EICIL: Joint Excitatory Inhibitory Cycle Iteration Learning for Deep Spiking Neural Networks[C]//Thirty-seventh Conference on Neural Information Processing Systems. 2023.

---

> ### Author Response · Authors · 2023-11-19
> **Response to Questions (part 1)**
>
> * **Could the authors provide a more comprehensive explanation of the excitation-inhibition mechanism and how it is used in this work? Although the authors keep mentioning it, the reviewer cannot understand how excitatory and inhibitory synapses are handled differently and how they are balanced.**
>
> *Theoretical analysis:*
>
> Firstly, the challenge of hybrid learning lies in the integration of global and local learning. While current hybrid learning methods have achieved excellent results in terms of accuracy[1], they could do slightly better in integrating the low energy consumption advantage of local learning. Therefore, based on the exciting Long-Term Potentiation (LTP) results of global learning and the inhibitory Long-Term Depression (LTD) results of local learning, an excitatory-inhibitory mechanism is adopted to balance the two.
>
> Secondly, in the cerebral cortex, the excitatory mechanism can enhance synaptic strength[2], while the inhibitory mechanism can weaken it[3,4]. Unlike previous works[5,6] that directly distinguish between excitation and inhibition at the synaptic and neuronal levels, we use the network connection status to differentiate between excitatory and inhibitory states.
>
> Finally, according to the excitatory-inhibitory mechanism, excitation and inhibition are automatically balanced. That is, when the network is overly excited, it should be inhibited, and when it is in an overly inhibited state, it should be excited. We propose an Excitatory-Inhibitory Hybrid Learning (EIHL) method, which better integrates the advantages of local and global learning, resulting in a model with high accuracy and low power consumption.
>
> > [1] Wu Y, Zhao R, Zhu J, et al. Brain-inspired global-local learning incorporated with neuromorphic computing[J]. Nature Communications, 2022, 13(1): 65.
>
> > [2] Malenka R C, Nicoll R A. Long-term potentiation--a decade of progress?[J]. Science, 1999, 285(5435): 1870-1874.
>
> > [3] Abraham W C, Bear M F. Metaplasticity: the plasticity of synaptic plasticity[J]. Trends in neurosciences, 1996, 19(4): 126-130.
>
> > [4] Li X, Steffens D C, Potter G G, et al. Decreased between‐hemisphere connectivity strength and network efficiency in geriatric depression[J]. Human brain mapping, 2017, 38(1): 53-67.
>
> > [5] Kern F B, Chao Z C. Short-term neuronal and synaptic plasticity act in synergy for deviance detection in spiking networks[J]. PLOS Computational Biology, 2023, 19(10): e1011554.
>
> > [6] Zhu G, Zhang Z, Zhang X Y, et al. Diverse Neuron Type Selection for Convolutional Neural Networks[C]//IJCAI. 2017: 3560-3566.

---

> ### Author Response · Authors · 2023-11-19
> **Response to Questions (part 2)**
>
> *Formula explanation:*
>
> We use the degree of network connectivity to distinguish between excitation and inhibition, so we set a target sparsity $b$ as the threshold for the network to be in an over-inhibited state. In local learning, due to the effect of LTD, the network sparsity gradually increases. When it exceeds the target sparsity $b$, the network is considered to be in an over-inhibited state, and the network will switch to excited global learning. The contraction formula Eq.6 accelerates the process of the network gradually becoming sparse in local learning.
>
> $\ \mathcal H(x) = thresh(b) \cdot x \cdot a ,\ \ \ \  a \in(0，1)， x \in \mathbb{N}   \ \ \ Eq.6$
>
> Actually, $\mathcal H(x)$  is a weight value, and contraction implies that weights less than the boundary value $\mathcal H(x)$  will shrink until they disconnect. Moreover, the target sparsity $b$ is a percentage, not a weight value, so $thresh(b)$ is the mapping of b on the weight distribution. In simpler terms, if the weights that are less than $thresh(b)$ in each layer are set to 0, the current sparsity will directly reach the target sparsity $b$. The network will immediately exhibit an over-inhibited state and will switch to excited global learning. However, defining the range and setting it to 0 directly is too crude and will cause a lot of unnecessary losses. Therefore, $x \cdot a$ is to give $\mathcal H(x)$ a slow expansion process from 0. $a$ first divides the $thresh(b)$ into multiple scales, and as $x$ gradually increases, $\mathcal H(x)$ will also slowly increase until it equals $thresh(b)$.
>
> However, we believe that directly setting the weights that are less than the boundary value $\mathcal H(x)$ to 0 is still crude. Therefore, we fix the update direction of the weights in the boundary value $\mathcal H(x)$ to only move towards 0, and then set the weights in the zero neighborhood to 0, to achieve the LTD result in a relatively smooth manner.
>
> $\ \mathcal{W}' = \mathcal{W} - \text{lr} \cdot \nabla \mathcal{W},\ \ \ \  \text{lr} > 0   \ \ \ Eq.7 $
>
> $\mathcal{W}$ denotes the weight that has not been updated, $\mathcal{W}'$ denotes the weight that has been updated, $\nabla W$ is the weight update amount of STDP and $\text{lr}$ denotes the learning rate.
>
> Summarize the contraction operation of EIHL on local learning, that is, The update direction of the weights within the $\mathcal H(x)$ boundary can only tend to 0, and the weight area close to 0 can be directly set to 0. Then $x++$ , $\mathcal H(x)$ gradually expands, and the final set to 0 area reaches the $thesh(b)$ area, the current sparsity reaches the target sparsity $b$, and switches to the excited global learning.
>
> *We hope that these revisions can help the reviewers to have a clearer understanding of our paper. If the reviewers have any questions or suggestions, please feel free to tell us, and we will improve them as soon as possible.*

---

> ### Author Response · Authors · 2023-11-23
> **Follow-up discussion**
>
> Dear Reviewer,
>
> Thank you sincerely for taking the time to review our work. We greatly appreciate your valuable feedback. If you have any further questions or concerns, we would be more than happy to address them promptly before the approaching deadline. Your input is crucial to improving the quality of our work.
>
> Alternatively, if you feel that the concerns you initially raised have been adequately addressed, we kindly request that you consider updating your evaluation to reflect this. Your updated evaluation would be immensely helpful to us.
>
> Once again, we extend our gratitude for your time and effort in reviewing our work. We look forward to your response.
>
> Thank You.
>
> Paper 557 Authors

---

> ### Comment · Reviewer_p6P6 · 2023-11-23
>
> I appreciate the great efforts the authors made in the rebuttal. Several concerns have been addressed.
>
> Now the description of the proposed method is clearer, and I understand the real meaning of excitation-inhibition mechanism in this work.
>
> Now I am conviced that the proposed method can achieve good sparsity. However, I still holds the idea that the overall performance is not good. Given the simplicity of the datasets and the effectiveness of the utilized network architectures, it is feasible to get similar accuracy and sparsity through straightforward regularization of a fine-tuned STDP model. Consequently, the true effectiveness of the proposed method requires further validation.
>
> Minor issue: the presentation of equations is still non-standard. For example, Eqns. 2 and 3 are like isolated items that do not belong to any sentences.
>
> Overall, in light of the addressed concerns and the efforts of the authors, I have revised my score from 3 to 5. Nevertheless, I maintain a relatively reserved evaluation.

---

### Official Review · Reviewer_K674 · 2023-10-28

**Soundness:** 3 good
**Presentation:** 3 good
**Contribution:** 3 good
**Rating:** 8
**Confidence:** 5

**Summary:**

This paper proposes a hybrid learning method that uses the neural excitation and inhibition mechanism to assist local learning and global learning (called EIHL), by simulating the biological neural excitation and inhibition mechanism to adjust the network connection state, thus integrating local learning and global learning. The experimental results also show that this method has advantages in accuracy and sparsity compared to separate local learning and global learning.

**Strengths:**

This paper is inspired by the biological neural excitation-inhibition mechanism and proposes a new hybrid learning method for SNN, which is more brain-like than previous methods and has originality. Moreover, this paper has some practical significance from both the biological perspective and the accuracy and sparsity of the experimental results. Furthermore, the whole paper is logically coherent and fluent, and the language is concise and clear.

**Weaknesses:**

The Fig.1 in this paper that explains the EIHL algorithm is too simple and not detailed enough, only showing the connection processing between the convolutional layer and the IF neuron layer. This figure could consider adding another layer to show the specific operation of the algorithm more finely.

**Questions:**

Is the EIHL method only applied to the convolutional layer? If not, I hope it can be reflected in the figure. I suggest updating and optimizing Fig.1.

---

> ### Author Response · Authors · 2023-11-19
> **Response to Weaknesses and Questions**
>
> We thank the reviewer for their feedback. We hope we have addressed their concerns and questions regarding the paper.
>
> * **The Fig.1 in this paper that explains the EIHL algorithm is too simple and not detailed enough, only showing the connection processing between the convolutional layer and the IF neuron layer. This figure could consider adding another layer to show the specific operation of the algorithm more finely.**
> * **Is the EIHL method only applied to the convolutional layer? If not, I hope it can be reflected in the figure. I suggest updating and optimizing Fig.1.**
>
> We apologize for the reviewer’s concern about Figure 1, we agree with the reviewer’s comments, and we have optimized Figure 1.
>
> Fig.1 is a description of the EIHL learning process. From the figure, it can be seen that after local learning, the synaptic connections weaken or even disconnect. And in global learning, the disconnected synapses can also reconnect and have the opportunity to learn again. Then, answer the reviewer’s questions about Fig.1: The EIHL method is applied to all layers that need to be trained. Because the main layer is the convolutional layer, only the convolutional layer is marked in the graph. But we thank the reviewer for pointing this out, and accept the reviewer’s opinion to refine Fig.1 to better present the EIHL algorithm.
>
> We are very grateful to the reviewers for their precise comments on this paper. We accept the reviewers’ suggestions and polish the paper accordingly, hoping to eliminate the reviewers’ concerns.

---

> > ### Comment · Reviewer_K674 · 2023-11-22
> >
> > Thank you for the detailed comments. The responses have addressed all my concerns and comments. After reading the other reviews, I still believe this paper presents a contribution worth of acceptance. Therefore, I would like to raise my score to 8.

---

> ### Author Response · Authors · 2023-11-23
> **Follow-up discussion**
>
> Dear Reviewer,
>
> We are very grateful for your favorable assessment of our work. Your choice to increase the score from 6 to 8 is very motivating and highly valued. We are thrilled to know that our efforts have been recognized and that our work has met your expectations.
>
> We also want to use this chance to ask if you have any further questions or doubts about our work. Your feedback is essential to us, and we want to make sure that we resolve any remaining problems before the final submission. We look forward to your response.
>
> Thank You.
>
> Paper 557 Authors

---

### Official Review · Reviewer_UpZe · 2023-11-04

**Soundness:** 3 good
**Presentation:** 3 good
**Contribution:** 3 good
**Rating:** 8
**Confidence:** 5

**Summary:**

This paper proposes an Excitation-Inhibition Mechanism assisted hybrid Learning (EIHL) algorithm. Inspired by the biological neural excitation-inhibition mechanism, it achieves adaptive adjustment of spiking neural network connectivity, and automatically alternates between global and local learning according to the growth or decay of synaptic strength which depends on the excitation-inhibition mechanism. It also conducts three experiments to demonstrate that this method has higher accuracy than global learning, and higher sparsity than local learning.

**Strengths:**

This paper proposes a new hybrid method of local and global learning, which is regulated by the biological neural excitation-inhibition mechanism. The paper also argues that the excitation-inhibition mechanism leads to sparse results for neural networks, which is an innovative perspective. Moreover, the language and logic of the introduction and method description are clear and smooth. Finally, the paper presents EIHL as a new point in the field of SNN training methods, and also provides new insights for ANN training methods, which has some significance.

**Weaknesses:**

The paper conducted three experiments to verify the performance advantages of the method, but the details of the third experiment are less described. The third experiment should specify which layer or the whole network is randomly pruned at different levels, and also explain the specific operation of random pruning.

**Questions:**

I would like to ask the author, is the random pruning at different levels applied to the whole network or to a specific layer? And will the synapses that are randomly cut off at the beginning be restored in the later learning or remain disconnected? Maybe it should be explained in the third experiment.

---

> ### Author Response · Authors · 2023-11-19
> **Response to Weaknesses and Questions**
>
> We thank the reviewer for their feedback, and we appreciate their recognition of the contribution of our work. We also apologize for the confusion that the reviewer still has. We hope that our answers can clear up the reviewer’s doubts.
>
> * **The paper conducted three experiments to verify the performance advantages of the method, but the details of the third experiment are less described. The third experiment should specify which layer or the whole network is randomly pruned at different levels, and also explain the specific operation of random pruning.**
> * **I would like to ask the author, is the random pruning at different levels applied to the whole network or to a specific layer? And will the synapses that are randomly cut off at the beginning be restored in the later learning or remain disconnected? Maybe it should be explained in the third experiment.**
>
> We apologize for the confusion caused by the insufficient description of the third experiment. This paper conducted three experiments in total: the first one is an evaluation of the sparsity and accuracy of EIHL, the second one is the influence of the contraction boundary value in EIHL, and the third one is a comparison of the performance of the model on different disconnect degrees.
>
> The different degrees of random pruning operation in the third experiment, it is performed after the weight initialization. For example, if the pruning degree is 20%, then the pruning object is all the weight parameters that need to be trained. The pruning step is performed layer by layer, and 20% of the weights are randomly selected and set to 0 in each layer, and their positions are recorded. In each subsequent gradient update, they and their gradients are set to 0. Therefore, the randomly selected weights will remain disconnected, which is why the sparsity and pruning degrees of STBP and STDP in Tab.3 are consistent.
>
> We are very grateful for the reviewer’s comments on this paper. We accept the reviewer’s opinions and have supplemented the description of the third experiment in the paper according to the above description. We hope that our explanation has eliminated the reviewer’s concerns.

---

> ### Author Response · Authors · 2023-11-23
> **Follow-up discussion**
>
> Dear Reviewer,
>
> We are very glad to receive your positive evaluation of our work. Your recognition of our efforts and the quality of our work means a lot to us.
>
> Meanwhile, we hope that our previous responses have addressed your questions and concerns. In addition, we would like to take this opportunity to inquire if you have any further questions or issues regarding our work. We value your feedback and we want to ensure that we have responded to all your comments and suggestions. If you have any additional queries or problems, please feel free to contact us before the final submission deadline.
>
> Thank You.
>
> Paper 557 Authors

---

### Official Review · Reviewer_iXWZ · 2023-11-09

**Soundness:** 2 fair
**Presentation:** 2 fair
**Contribution:** 2 fair
**Rating:** 5
**Confidence:** 3

**Summary:**

The authors proposed an Excitation-Inhibition Mechanism-assisted Hybrid Learning (EIHL) algorithm for training Spiking Neural Networks (SNNs), a learning algorithm that hybrid local learning rule and global learning rule.
Experiments on CIFAR10/100 and DVS-CIFAR10 showed that EIHL outperforms other methods in terms of accuracy and sparsity.

**Strengths:**

1. Fusion of Global and Local Learning Rules: According to the authors, the integration of both global and local learning paradigms could potentially pave the way for attaining enhanced performance and energy efficiency in neural networks.

2. Performance on CIFAR Benchmark: The authors successfully demonstrated an improvement in performance on the CIFAR dataset when compared to the traditional backpropagation technique.

**Weaknesses:**

1. The authors should elucidate their contributions more explicitly in order to provide a comprehensive understanding of the research.

2. In the context of hybrid learning, 'STDP' process employs a contraction curve to facilitate Long-Term Depression. Nevertheless, the authors have not adequately expounded upon the association between LTD and STDP, and the proposed method do not have a dependence of the spike timing. It's not clear why excitation should be like STBP and depression should be like STDP.

3. The accuracy of references should be ensured. For example, the paper states, "Spike-Timing Dependent Plasticity (STDP) was proposed based on these rules by Caporale & Dan (2008)." However, the discovery of STDP predates 2008. Caporale & Dan (2008) is a review paper.

**Questions:**

1. In the context of the hybrid learning rule, what is the significance of excitatory and inhibitory synapses, given that STDP and STBP do not appear to rely on the distinction between these synapse types? Furthermore, it seems that excitatory and inhibitory synapses are not typically delineated in deep spiking neural networks.

2. Weight pruning is a technique employed in deep learning to increase network sparsity by eliminating the smallest weights. Please elucidate the distinctions between the 'STDP' process in EIHL and weight pruning techniques in deep learning.

3. Kindly review the terminology and references utilized in the manuscript to ensure a more precise and coherent presentation of the research study.

---

> ### Author Response · Authors · 2023-11-19
> **Response to Weaknesses**
>
> We appreciate the feedback from the reviewer. We hope our subsequent responses have adequately addressed their concerns and questions regarding the paper.
>
> **Weakness Ⅰ:**
>
> * **The authors should elucidate their contributions more explicitly in order to provide a comprehensive understanding of the research.**
>
> We apologize for the confusion, and then according to the reviewer’s suggestion, we re-summarized the contributions of this work:
>
> 1. We propose an excitation-inhibition mechanism-assisted hybrid learning(called EIHL), which can combine the high accuracy of global learning and the low energy consumption of local learning and showed excellent accuracy and sparsity in the experiments on three datasets and two models.
> 2. We obtained inspiration from neuroscience and adopted the excitation-inhibition mechanism to solve the problem of how to reasonably integrate hybrid learning. The experimental results showed that EIHL also achieved sparsity advantages, which made it possible to deploy on hardware.
> 3. We used the neural excitation-inhibition mechanism to achieve the integration of global and local learning and implemented it by adjusting the weights. This not only provides a new perspective for the field of SNN training methods but also prepares for the generalization of EIHL to the ANN domain.
>
> **Weakness Ⅱ:**
>
> * **In the context of hybrid learning, 'STDP' process employs a contraction curve to facilitate Long-Term Depression. Nevertheless, the authors have not adequately expounded upon the association between LTD and STDP,**
>
> Spike-Timing Dependent Plasticity (STDP)[1] is a synaptic plasticity mechanism that depends on the firing sequence of pre- and post-synaptic neurons. In STDP, if the pre-synaptic neuron fires before the post-synaptic neuron, synaptic strength increases; if the pre-synaptic neuron fires after the post-synaptic neuron, synaptic strength decreases. Long-Term Potentiation (LTP) and Long-Term Depression (LTD) are the two components that constitute STDP[2]. However, significant LTP only occurs at synapses with relatively low initial strength, whereas the extent of LTD does not show an obvious dependence on the initial synaptic strength[3]. Therefore, the overall effect of STDP tends to exhibit LTD.
>
> * **and the proposed method do not have a dependence of the spike timing.**
>
> Our approach does not rely on spiking timing, yet the original intention of Spiking Neural Networks (SNN) is to emulate brain systems, distinguishing it from Artificial Neural Networks (ANN). Therefore, our process of addressing the hybrid learning problem through biological inspiration aligns with the SNN field, and there is no necessity to utilize spiking timing. Lastly, it is precisely because we do not depend on spiking timing that EIHL can be conveniently extended to the ANN field.
>
> * **It's not clear why excitation should be like STBP and depression should be like STDP.**
>
> Due to BP’s powerful learning capabilities, most weights increase after STBP learning, which corresponds to the excitation of LTP[4]. STDP, on the other hand, exhibits an overall LTD effect, corresponding to inhibitory LTD[4].
>
> **Weakness Ⅲ:**
>
> * **The accuracy of references should be ensured. For example, the paper states, "Spike-Timing Dependent Plasticity (STDP) was proposed based on these rules by Caporale & Dan (2008)." However, the discovery of STDP predates 2008. Caporale & Dan (2008) is a review paper.**
>
> First of all, we would like to thank the reviewer for pointing out the mistake about the STDP reference, and we apologize for this mistake. We have corrected the STDP reference in section 2.1 to “Based on these rules, spike-timing dependent plasticity (STDP) was proposed by Song et al. (2000)[1].” We appreciate the reviewer’s correction.
>
> *We hope that these revisions can help the reviewers to have a deeper understanding and a clearer understanding of the contributions and innovations of our paper. If the reviewers have any questions or suggestions, please feel free to tell us, and we will improve them as soon as possible.*
>
> > [1] Song S, Miller K D, Abbott L F. Competitive Hebbian learning through spike-timing-dependent synaptic plasticity[J]. Nature neuroscience, 2000, 3(9): 919-926.
>
> > [2] Zenke F, Gerstner W, Ganguli S. The temporal paradox of Hebbian learning and homeostatic plasticity[J]. Current opinion in neurobiology, 2017, 43: 166-176.
>
> > [3] Bi G, Poo M. Synaptic modifications in cultured hippocampal neurons: dependence on spike timing, synaptic strength, and postsynaptic cell type[J]. Journal of neuroscience, 1998, 18(24): 10464-10472.
>
> > [4] Abraham W C, Bear M F. Metaplasticity: the plasticity of synaptic plasticity[J]. Trends in neurosciences, 1996, 19(4): 126-130.

---

> ### Author Response · Authors · 2023-11-19
> **Response to Questions Ⅰ**
>
> We appreciate the feedback from the reviewer. We hope our subsequent responses have adequately addressed their concerns and questions regarding the paper.
>
> **Question Ⅰ:**
>
> * **In the context of the hybrid learning rule, what is the significance of excitatory and inhibitory synapses, given that STDP and STBP do not appear to rely on the distinction between these synapse types?**
>
> Firstly, the challenge of hybrid learning lies in the integration of global and local learning. While current hybrid learning methods have achieved excellent results in terms of accuracy[1], they could do slightly better in integrating the low energy consumption advantage of local learning. Therefore, based on the exciting Long-Term Potentiation (LTP) results of Spike-Timing-Dependent Plasticity (STDP) and the inhibitory Long-Term Depression (LTD) results, an excitatory-inhibitory mechanism is adopted to balance the two.
>
> Secondly, in the cerebral cortex, the excitatory mechanism can enhance synaptic strength[2], while the inhibitory mechanism can weaken it[3,4]. Unlike previous works[5,6] that directly distinguish between excitation and inhibition at the synaptic and neuronal levels, we use the network connection status to differentiate between excitatory and inhibitory states.
>
> Finally, according to the excitatory-inhibitory mechanism, excitation and inhibition are automatically balanced. That is, when the network is overly excited, it should be inhibited, and when it is in an overly inhibited state, it should be excited. We propose an Excitatory-Inhibitory Hybrid Learning (EIHL) method, which better integrates the advantages of local and global learning, resulting in a model with high accuracy and low power consumption.
>
> * **Furthermore, it seems that excitatory and inhibitory synapses are not typically delineated in deep spiking neural networks**
>
> For this reason, this paper does not adopt the approach of explicitly distinguishing between excitatory and inhibitory synapses in the network. Instead, it differentiates between excitation and inhibition based on the connection status of the network.
>
> > [1] Wu Y, Zhao R, Zhu J, et al. Brain-inspired global-local learning incorporated with neuromorphic computing[J]. Nature Communications, 2022, 13(1): 65.
>
> > [2] Malenka R C, Nicoll R A. Long-term potentiation--a decade of progress?[J]. Science, 1999, 285(5435): 1870-1874.
>
> > [3] Abraham W C, Bear M F. Metaplasticity: the plasticity of synaptic plasticity[J]. Trends in neurosciences, 1996, 19(4): 126-130.
>
> > [4] Li X, Steffens D C, Potter G G, et al. Decreased between‐hemisphere connectivity strength and network efficiency in geriatric depression[J]. Human brain mapping, 2017, 38(1): 53-67.
>
> > [5] Kern F B, Chao Z C. Short-term neuronal and synaptic plasticity act in synergy for deviance detection in spiking networks[J]. PLOS Computational Biology, 2023, 19(10): e1011554.
>
> > [6] Zhu G, Zhang Z, Zhang X Y, et al. Diverse Neuron Type Selection for Convolutional Neural Networks[C]//IJCAI. 2017: 3560-3566.

---

> ### Author Response · Authors · 2023-11-19
> **Response to Questions Ⅱ and Ⅲ**
>
> We appreciate the feedback from the reviewer. We hope our subsequent responses have adequately addressed their concerns and questions regarding the paper.
>
> **Question Ⅱ:**
>
> * **Weight pruning is a technique employed in deep learning to increase network sparsity by eliminating the smallest weights. Please elucidate the distinctions between the 'STDP' process in EIHL and weight pruning techniques in deep learning.**
>
> We apologize for the confusion regarding the distinctions between the ‘STDP’ process in EIHL and weight pruning techniques in deep learning.
>
> Firstly, weight pruning techniques in deep learning are more of an engineering technique, aimed at reducing the complexity and computation of the model. However, the ‘STDP’ process in EIHL is part of the hybrid training method, aimed at mimicking LTD to suppress network excitation.
>
> Secondly, weight pruning techniques in deep learning do not fully simulate the working principle of biological neural systems, but the ‘STDP’ process in EIHL is a form of synaptic plasticity. It is just that the concept of pruning has some similarities with synaptic plasticity.
>
> Finally, in terms of the specific operation, the ‘STDP’ process in EIHL and weight pruning techniques in deep learning have some similarities. Similarity: They are both implemented by removing the smallest weights. Difference: Pruning is to directly set the weights that are less than a certain value or a certain percentage of the weight distribution to zero. The ‘STDP’ process in EIHL is to fix the update direction of the weights that are less than H(x) according to Eq.7, and then set the weights in the zero domain to zero. Moreover, the proportion of weights that need to be shrunk in the ‘STDP’ process in EIHL is automatically adjusted according to the increase of the target sparsity b, not artificially specified.
>
> **Question Ⅲ:**
>
> * **Kindly review the terminology and references utilized in the manuscript to ensure a more precise and coherent presentation of the research study.**
>
> We are very grateful for the reviewer’s comments, and we have carefully checked and polished our paper, correcting the citation errors and incoherent parts. We have tried our best to express the paper more accurately and coherently.
>
> *We hope that these revisions can help the reviewers to have a clearer understanding of our paper. If the reviewers have any questions or suggestions, please feel free to tell us, and we will improve them as soon as possible.*

---

> ### Author Response · Authors · 2023-11-23
> **Follow-up discussion**
>
> Dear Reviewer,
>
> Thank you so much for your review -- please let us know if you have any remaining questions or concerns so that we can address them before the deadline coming soon. Alternatively, if you feel that your original concerns are addressed, we would appreciate updating your evaluation to reflect that.
>
> Thank You.
>
> Paper 557 Authors

---

> ### Comment · Reviewer_iXWZ · 2023-11-23
> **Thank you for your comment**
>
> I have carefully read the authors' rebuttal and while I acknowledge their efforts to address the concerns raised. However, I remain unconvinced that the contributions presented in the paper are significant enough to warrant a change in my evaluation. STDP is a important part of EIHL, however the 'STDP' training algorithm in EIHL is not STDP,  since it does not dependent on spike time and contradict to the defination of STDP. I encourage the authors to be careful of the basic concepts in neuroscience. Therefore, I will maintain my original score for the paper.

---

### Meta-Review · Area_Chair_meFj · 2023-12-11

**Metareview:**

This paper proposes a novel hybrid learning method with neural excitation and inhibition mechanisms which integrates local learning and global learning rules.  The experimental results show that the proposed method has advantages in accuracy and sparsity compared to previous methods.  Although there was a moderately strong difference in opinion among the reviewers about the paper's overall contribution, the enthusiasm of the 2 more positive reviewers outweighed the more borderline appraisals of the 2 more negative reviewers, and I believe the paper should be accepted to ICLR.  Congratulations!  Please revise the manuscript to address all reviewer comments and questions.

**Justification For Why Not Higher Score:**

Two of the 4 reviewers felt the paper made only a slight advance over previous results, and thus it seems hard to justify a spotlight or oral presentation.

**Justification For Why Not Lower Score:**

Two reviewers gave highly enthusiastic reviews, and their enthusiasm seemed to outweigh the two more negative reviews.  Moreover, the more negative reviewers did not engage very much with the rebuttals, so I am inclined to side with the authors (who wrote very detailed reviews).

---

### Decision · Program_Chairs · 2024-01-16

Accept (poster)